# Exploring Microbial Ecosystem Services for Environmental Stress Amelioration: A Review

**DOI:** 10.3390/ijms26104515

**Published:** 2025-05-09

**Authors:** Pradeep Semwal, Anand Dave, Juveriya Israr, Sankalp Misra, Manish Kumar, Diby Paul

**Affiliations:** 1Microbial Technologies Division, Council of Scientific and Industrial Research-National Botanical Research Institute (CSIR-NBRI), Rana Pratap Marg, Lucknow 226001, Uttar Pradesh, India; semwalpradeep640@gmail.com; 2Academy of Scientific and Innovative Research (AcSIR), Ghaziabad 201002, Uttar Pradesh, India; 3Department of Microbiology, Seventh Day Adventist Arts and Science College, Ahmedabad 380008, Gujrat, India; ananddave16@gmail.com; 4Faculty of Biosciences, Institute of Biosciences and Technology, Shri Ramswaroop Memorial University, Lucknow-Deva Road, Barabanki 225003, Uttar Pradesh, India; juveriyaisrar2016@gmail.com (J.I.); sankalpnbri@gmail.com (S.M.); 5Amity Institute of Biotechnology, Amity University Gwalior, Gwalior 474005, Madhya Pradesh, India; mkumar@gwa.amity.edu; 6College of Life and Health Sciences, Truett McConnel University, Cleveland, GA 30528, USA

**Keywords:** PGPR, abiotic stress management, environmental sustainability, recalcitrant compounds, xenobiotics

## Abstract

The increasing global population and intensifying resource limitations present a formidable challenge for sustainable crop production, especially in developing regions. This review explores the pivotal role of microbial ecosystem services in alleviating environmental stresses that impede agricultural productivity. Soil microbiota, particularly plant growth-promoting microbes (PGPMs), are integral to soil health and fertility and plant resilience against both abiotic (drought, salinity, temperature extremes, heavy metals) and biotic (pathogen) stresses. These microorganisms employ a variety of direct and indirect mechanisms, including the modulation of phytohormones, nutrient solubilization, the production of stress-alleviating enzymes, and the synthesis of antimicrobial compounds, to enhance plant growth and mitigate adverse environmental impacts. Advances in microbial biotechnology have expanded the toolkit for harnessing beneficial microbes, enabling the development of microbial inoculants and consortia tailored for specific stress conditions. This review highlights the multifaceted contributions of soil microbes, such as improving nutrient uptake, promoting root development, facilitating pollutant degradation, and supporting carbon sequestration, all of which underpin ecosystem resilience and sustainable agricultural practices. Furthermore, the synergistic interactions between plant roots and rhizospheric microbes are emphasized as key drivers of soil structure enhancement and long-term productivity. By synthesizing current research on the mechanisms of microbe-mediated stress tolerance, this review underscores the potential of microbial interventions to bridge the gap between food security and environmental conservation. The integration of microbial solutions into agroecosystems offers a promising, eco-friendly strategy to revitalize soils, boost crop yields, and ensure agricultural sustainability in the face of mounting environmental challenges.

## 1. Introduction

Soil health is fundamentally shaped by its microbial biomass, which is essential for sustaining life and ensuring crop productivity [1]. Sustainable agriculture aims to enhance yields with minimal environmental impact, a goal challenged by various abiotic and biotic stresses [2]. Soil microbes, as natural colonizers, play a vital role in maintaining soil health and developing stress tolerance [3]. The diversity of soil microbiota directly influences soil fertility, as plant–microbe interactions and related processes improve the soil’s physico-chemical properties and boost plant productivity [4]. The rhizosphere soil region surrounding plant roots serves as a dynamic interface for root–microbe communication, contributing to soil health and plant growth [5].

The rhizospheric microbes that contribute to plant productivity and soil health are broadly known as plant growth-promoting microbes (PGPMs), helping plants by facilitating improved nutrient bioavailability, nutrient uptake, and the production of plant growth-promoting substances [6]. Tolerance against biotic stress, such as pathogens, and abiotic stress, such as drought, salinity, extreme temperatures, and pollution, is a crucial attribute that soil microbes contribute to plants [7]. Different mechanisms have been adopted by various functionally and genetically diverse microbes that inhabit soil, including groups such as bacteria, fungi, and actinomycetes [8].

Microbes, other than being a biological component of soil, influence various biogeochemical cycles that help in the recycling of nutrients, therefore enhancing its mineral composition. Several microbes have been known to have a role in plant growth promotion either directly or indirectly, such as through biocontrol activity, i.e., the suppression of plant pathogens and the production of certain substances that stimulate the growth of plants; through their role in bioremediation, i.e., the removal of pollutants; or through their stress tolerance abilities, which may help plants dodge various abiotic stresses [9,10]. Since microbes respond quickly to environmental changes, they could also be an excellent means for evaluating as well as improving plant growth through different pathways, which could be direct or indirect. This includes modulating phytohormone signaling, preventing plant pathogens, and enhancing the availability and uptake of nutrients by plants [11].

This review comprehensively examines the diverse mechanisms by which soil microbes contribute to the amelioration of environmental stresses in agroecosystems and discusses the broader implications of microbial interventions for sustainable agriculture and environmental health. Initially, this review provides an overview of the major abiotic stresses affecting crop productivity and soil health, followed by an in-depth discussion of the roles and mechanisms of plant growth-promoting microbes (PGPMs) in stress mitigation. The subsequent sections explore the biotechnological approaches for harnessing microbial diversity, case studies demonstrating successful microbial applications, and the challenges and prospects in integrating microbial solutions into mainstream agricultural practices. By systematically addressing these aspects, this review aims to offer a holistic perspective on the potential of microbial ecosystem services in supporting resilient and sustainable food production systems.

## 2. Abiotic Stress and Its Impact on Plant Productivity and Soil Health

Plants require optimal conditions, including nutrients, water, light, and temperature, for healthy growth, while extremes or deficiencies in these factors lead to stress and reduced productivity [12,13,14]. Abiotic stresses such as drought, salinity, temperature extremes, nutrient deficiency, and pollution disrupt plant metabolism and soil health, affecting soil structure and plant biochemical responses [15,16,17,18]. Under water stress, plants show reduced growth, smaller leaves and seeds, and delayed flowering; heat stress triggers protein repair and antioxidant responses; and salt stress disrupts ion balance, suppresses germination, and accelerates senescence [19,20,21,22,23]. Low temperatures slow metabolic reactions, while heavy metals damage proteins and reduce nutrient uptake [24,25]. Moreover, adaptations to abiotic stress can sometimes increase susceptibility to biotic stresses, as seen with reactive oxygen species production, making plants more susceptible to certain pathogens [26,27,28].

## 3. Role of Microbes in Abiotic Stress Management

A diverse range of plant growth-promoting rhizobacteria (PGPRs), including genera such as *Bacillus*, *Pseudomonas*, *Ochromobactrum*, *Enterobacter*, *Alcaligenes*, and *Jeotgalicoccus*, have demonstrated their ability to enhance plant growth and productivity under various abiotic stress conditions [22,29,30,31,32]. These bacteria facilitate plant resilience through mechanisms like phosphate solubilization, indole-3-acetic acid (IAA) production, siderophore synthesis, and 1-aminocyclopropane-1-carboxylic acid (ACC) deaminase activity, collectively enhancing nutrient acquisition and reducing ethylene-induced stress in plants [33]. Mycorrhizal associations also significantly contribute to stress amelioration by improving nutrient uptake. Vesicular Arbuscular Mycorrhizae (VAM), through their extensive hyphal networks, enlarge the root surface area for nutrient absorption and enhance the bioavailability of phosphorus and micronutrients, crucial under nutrient-deficient or stress-prone soils [34]. Empirical studies have substantiated the role of PGPRs in heavy metal remediation and stress tolerance. *Pseudomonas* spp. have demonstrated potential in mitigating the toxic effects of heavy metals like Nickel (Ni), Zinc (Zn), and Copper (Cu), with reported increases in plant biomass and tolerance to water stress through multiple mechanisms, including phosphate solubilization and IAA production [33]. In field experiments, arsenite- and arsenate-tolerant bacterial consortia significantly improved rice growth and arsenic detoxification, enhancing root and shoot biomass, and reducing arsenic accumulation in the edible parts [10].

Regarding *Bacillus* species, extensive experimental data reinforce their functional role in abiotic stress mitigation. For instance, *Bacillus pumilus* has been observed to confer salt and drought tolerance in potato crops through increased activity of stress-related enzymes such as catalase and peroxidase, improved chlorophyll content, and enhanced root biomass in both greenhouse and field trials [35]. Similarly, *Bacillus amyloliquefaciens* inoculation has shown significant improvements in drought resistance and root morphology in rice, supporting enhanced water uptake and growth under deficit irrigation regimes [36]. In a separate study, *Bacillus altitudinis* NKA32, an endophytic strain with ACC deaminase activity, ameliorated salt stress in rice through the modulation of osmolyte accumulation, antioxidant enzyme activation, and the altered expression of stress-responsive genes [36].

Further studies have highlighted the metabolic versatility of PGPR strains. The genomic characterization of salt-tolerant *Bacillus* and *Jeotgalicoccus* strains revealed diverse metabolic pathways associated with the osmotic stress response, compatible solute biosynthesis, and heavy metal detoxification [22]. *Jeotgalicoccus huakuii* NBRI 13E, in particular, showed enhanced growth promotion in saline soils, boosting maize root and shoot biomass and improving the chlorophyll stability index under salt stress [31]. Additionally, members of *Nocardia*, *Hydrogenomonas*, *Xanthomonas*, *Microbacterium*, *Flavobacterium*, and *Pseudomonas* have shown significant roles in the degradation of hydrocarbons and xenobiotic compounds, alleviating stress caused by environmental pollutants [37]. These microbes utilize co-metabolic degradation, bioleaching, and bioaccumulation strategies to reduce the bioavailability and toxicity of recalcitrant compounds, ultimately promoting plant health.

In heavy-metal-contaminated soils, PGPRs such as *Ochrobactrum* sp. NBRISH6 have been experimentally shown to improve maize yield, enhance soil enzyme activity (dehydrogenase and phosphatase), and stimulate host plant immune responses, demonstrating a dual role in soil remediation and productivity enhancement [30]. Overall, microbial inoculants represent an effective strategy for mitigating multiple abiotic stresses, with growing experimental support from field trials, metabolomic profiling, and enzyme activity assays underscoring their promise in sustainable agriculture.

## 4. Mechanism Involved in Microbe-Mediated Abiotic Stress Mitigation

PGPMs follow direct or indirect mechanisms for conferring stress tolerance to plants. Direct mechanisms include the facilitation of essential macro- and micro-nutrients along with other different processes, such as (1) the modulation of phytohormone levels; (2) the production of molecules such as siderophores in the rhizospheric zone or the induced synthesis of aminocyclopropane-1-carboxylate deaminase, which is known to decrease ethylene levels; (3) altering the osmotic state of cells; (4) an enhanced ability to solubilize phosphorus, etc. [38,39]. The production of metabolites with antimicrobial potential is included under the indirect stress tolerance mechanism. This includes the induced biosynthesis of antimicrobial compounds such as hydrogen cyanate, 2,4-diacetyl phloroglucinol, tensins, phenazine, pyoluteorin, pyrrolnitrin, viscosinamide, etc. Such molecules are known to inhibit phytopathogens, thus acting as a means of biotic stress tolerance [40]. Different microbes may show their own set of strategies to develop stress tolerance, such as rhizospheric microbe *Pseudomonas* spp.; the synthesis of carboxylates that can chelate certain nutrients affects nutrient bioavailability [41]. *Pseudomonas* spp. also may induce the production of substances that aid in dissolving mineral oxides, such as phenazines, riboflavin, quinines, and humic substances [42].

An ideal PGPR has all the essential properties to enhance plant growth and soil health through its high rhizosphere competence, high multiplicity, broad-spectrum anti-pathogen action, and nutrient-providing abilities [43]. They play an important role in the mineralization of many elements, which results in increased soil nutrient content and improved soil enzymatic activity. As a result, increased nutrient availability assists in plant growth. For example, the solubilization of phosphorus is facilitated by organic acids produced by rhizospheric microbes. Enzymes such as phosphatases utilize organic phosphorus as a substrate and convert it into inorganic form [44]. Similarly, phytases release phosphorus from phytic acid, which is the organic form of phosphorus, mainly present in soil. Developing these bacteria as inoculants could be an effective strategy for developing greater nutrient mass in the soil, leading to improved crop yield [45]. Potassium, an important element required for cell growth regulation, is found to be insufficient in most natural settings. An insoluble form of potassium from the soil can be used by potassium-solubilizing rhizobacteria to convert it into a soluble form. This is achieved through the production of organic acids such as acetate, citrate, oxalate, etc. [46]. Zinc, another important micronutrient, is present in insoluble form in the soil. Zinc-solubilizing bacteria can convert insoluble zinc into its divalent cation (soluble form) either through the production of acids such as carbonic acid and nitric acid [47], or by chelating compounds known as siderophores. These bacteria, with the ability to mobilize zinc complexes in the soil, are efficient mineralizers and can improve soil as well as plant health [48].

Many PGPRs can stimulate the synthesis of phytohormones, thus promoting their growth and development. IAA producing bacteria such as *Pseudomonas*, *Rhizobium*, *Agrobacterium*, and *Bacillus* species promote cell growth and ameliorate metal stress through their metal sorption ability [6]. Similarly, microbes that can produce 1-aminocyclopropane-1-carboxylate deaminase (ACCD) can decrease ethylene levels, thus increasing tolerance to stress (Figure 1). Moreover, ACC is catalyzed into α-ketobutyrate and ammonium and acts as a source of carbon and nitrogen [49]. Similarly, PGPMs can produce Volatile Organic Compounds (VOCs) that have direct and indirect roles in plant growth promotion as well as soil health. For example, HCN-producing *Bacillus* and *Pseudomonas* species show biocontrol activity against many potential phytopathogens and also have an important role in the geochemical processing of soil substrata through metal chelation and the solubilization of elements such as phosphate [50].

Several rhizospheric microbes that exhibit plant growth-promoting (PGP) traits can be effectively utilized for the production of biofertilizers. The bioformulations can have attributes such as (1) nitrogen-fixing ability, (2) phosphate solubilization, (3) ACC deaminase activity, and (4) the accumulation of heavy metals and xenobiotics along with other PGP attributes [5,37]. The microbial formulations can lead to products such as biofertilizers for enhanced nutrient supply, biopesticides for preventing plant pathogens, biostimulants for improving nutrient availability and uptake, etc. Today, these products are the key contributors to sustainable agriculture and are an integral part of the agroecosystem [51]. A polymicrobial formulation developed from a consortium of microbes could also be developed with microbial combinations that have complementary plant growth-promoting attributes [52]. However, the efficacy of these formulations will depend upon how synergistic the interaction of microbes is, as well as upon the complementary functionality of the microbes. However, genes responsible for stress amelioration properties can be identified, and the information generated can be utilized for stress management for crops [53,54]. Native soil microbes can also be genetically engineered for picking specific stress amelioration conditions, including strategies such as gene silencing, and the upregulation of particular regulons can be carried out. For example, *Pseudomonas korensis* has shown heavy metal bioremediation ability in *Miscanthus sinensis* through ACC deaminase and IAA production [55].

## 5. Biodegradation of Recalcitrant Compounds

Recalcitrant compounds, often classified as persistent organic pollutants (POPs), are notorious for their resistance to natural degradation due to their stable, complex molecular structures [56]. These chemical substances do not easily break down in the environment and can persist for decades or even centuries, posing significant ecological and health risks. Common examples of refractory pollutants include polychlorinated biphenyls (PCBs), polycyclic aromatic hydrocarbons (PAHs), and various synthetic pesticides such as dichlorodiphenyltrichloroethane (DDT) and chlordane [57]. These compounds were widely used in industrial processes, agriculture, and pest control before their hazardous nature was fully understood. Today, they are recognized as major environmental contaminants due to their toxicity, persistence, and tendency to bioaccumulate in the food chain [58].

The persistence of recalcitrant compounds is primarily attributed to their molecular stability, which makes them resistant to conventional chemical or physical breakdown processes (Figure 2). For instance, PCBs have a biphenyl structure with chlorine atoms attached, making them hydrophobic and resistant to hydrolysis, oxidation, or photodegradation [59]. Similarly, PAHs consist of multiple fused aromatic rings that provide exceptional stability against microbial or chemical attacks. This stability enables these compounds to accumulate in soils, sediments, and aquatic systems, leading to long-term contamination and serious ecological disruptions [60]. Once introduced into the environment, these substances tend to adhere to organic matter, making their removal even more challenging.

The biodegradation process is often facilitated by microbial consortia—complex communities of microorganisms that work in a coordinated manner to degrade recalcitrant pollutants [61]. These consortia include bacteria, fungi, and archaea with complementary metabolic capabilities, allowing for the more efficient and complete breakdown of contaminants. For example, species of *Pseudomonas* and *Bacillus* are well documented regarding their roles in degrading aromatic hydrocarbons, leveraging their versatile enzymatic systems to initiate and propagate the breakdown of these compounds [62,63]. Fungal species like Saccharomyces and other yeast strains contribute significantly to the degradation process by secreting extracellular enzymes that attack large and complex molecules. The biodegradation process can vary significantly based on factors such as the nature of the pollutant, the environmental conditions (pH, temperature, oxygen levels), and the composition of the microbial community [64].

Enzymatic degradation involves several key reactions, including oxidation, reduction, hydrolysis, and dehalogenation, which collectively work to break down complex organic molecules into simpler ones [65]. For example, enzymes like monooxygenases and dioxygenases introduce oxygen atoms into PAH molecules, initiating the breakdown of their aromatic rings. Similarly, dehalogenases are essential for removing halogen atoms from PCBs and other chlorinated compounds, converting them into less harmful intermediates [66]. These enzyme-mediated reactions often result in the formation of intermediate metabolites, which are simpler chemical entities that can be further degraded or assimilated by microbial cells [67]. Advanced biotechnological techniques, including metagenomics and synthetic biology, have revolutionized the field of microbial biodegradation. Metagenomics enables the study of microbial communities directly from environmental samples without the need for culturing, revealing the genetic diversity and metabolic potential of these microbes [68]. This approach allows researchers to identify novel genes and enzymes involved in the degradation pathways of recalcitrant compounds, facilitating the development of more targeted bioremediation strategies [69]. The data gathered from metagenomic analyses provide insights into the microbial interactions and environmental factors that enhance biodegradation efficiency.

Synthetic biology takes this knowledge a step further by engineering microorganisms with enhanced capabilities to degrade specific pollutants. These genetically modified organisms (GMOs) are designed to express optimized enzymes or metabolic pathways that accelerate the breakdown of recalcitrant compounds [70]. The integration of these engineered microbes into contaminated environments has shown promising results in enhancing the rate and extent of pollutant degradation [71]. Additionally, the stability and activity of these engineered microorganisms can be fine-tuned to withstand harsh environmental conditions, thereby increasing their survival and functional efficiency in field applications [72]. This approach also allows the manipulation of microbial consortia by introducing engineered strains that complement the natural community’s capabilities, thereby creating a more robust bioremediation system. The ability to manipulate and enhance microbial pathways holds great promise in addressing the limitations of traditional biodegradation methods and providing sustainable solutions for the remediation of contaminated environments.

## 6. Biodegradation of Xenobiotics

Xenobiotics refer to synthetic compounds introduced into the environment that do not naturally occur in nature. These substances encompass a wide range of industrial chemicals, pharmaceuticals, pesticides, herbicides, and by-products resulting from human activities like manufacturing and agricultural processes. Due to their synthetic origin and complex chemical structures, xenobiotics are often resistant to natural degradation processes and can persist in soil, water, and air, posing substantial risks to both environmental and human health [36,73,74]. Examples of these substances include industrial solvents, plasticizers, flame retardants, dyes, and several pharmaceutical residues. Their resistance to breakdown means that once released into the environment, they can accumulate and exert toxic effects on ecosystems and organisms, impacting biodiversity and ecological balance [75].

One of the primary concerns with xenobiotics is their potential to cause long-term ecological disruptions. Their presence in water bodies can lead to the contamination of aquatic systems, while their persistence in soils can affect soil fertility and plant growth [76]. Over time, these compounds can bioaccumulate in living organisms, move up the food chain, and ultimately enter human diets, raising concerns about chronic health issues like endocrine disruption, carcinogenicity, and mutagenicity. The accumulation of xenobiotics in the environment highlights the need for effective remediation strategies to reduce their adverse impacts.

Microbial biodegradation represents a natural and promising approach to mitigating the harmful effects of xenobiotics in the environment. Microbes possess the remarkable ability to utilize xenobiotics as sources of carbon and energy, transforming them into less toxic or non-toxic compounds [77]. This transformation occurs through metabolic processes that involve a series of enzymatic reactions capable of breaking down the complex molecular structures of xenobiotics into simpler substances [78]. The metabolic diversity of these microorganisms allows them to target different types of xenobiotics, enhancing their adaptability to various environmental conditions [79,80].

Among the most effective microorganisms in xenobiotic degradation are species like *Pseudomonas*, *Rhodococcus*, and *Mycobacterium*, known for their extensive enzymatic repertoire that includes oxygenases, dehalogenases, hydrolases, and reductases [79]. These enzymes play a crucial role in initiating the breakdown of stable chemical bonds within xenobiotics, facilitating their conversion into intermediates that can be further degraded into harmless end products [81]. For example, oxygenases introduce oxygen molecules into xenobiotic compounds, leading to ring cleavage in aromatic compounds and promoting their further breakdown into simpler organic acids [63].

The degradation of polychlorinated biphenyls, a form of xenobiotics notorious for their environmental persistence and toxic effects, has been extensively studied in microbial systems. PCB degradation can occur through both aerobic and anaerobic pathways, with bacteria such as *Pseudomonas putida* playing a significant role. In aerobic environments, microorganisms utilize oxygen-dependent enzymes like biphenyl dioxygenases to initiate the oxidation of PCB molecules, transforming them into chlorobenzoic acids, which are less toxic and more easily biodegradable (Table 1). This process is further supported by subsequent enzymatic steps that break down the intermediate products into benign substances like carbon dioxide and water. The anaerobic degradation of PCBs, on the other hand, relies on the process of reductive dechlorination, wherein certain bacteria utilize PCBs as electron acceptors, stripping off chlorine atoms in the absence of oxygen (Table 1). This stepwise dechlorination significantly reduces the toxicity of PCBs, creating compounds that are more amenable to complete mineralization under aerobic conditions. The synergy between anaerobic and aerobic microbial communities enables a more comprehensive degradation process, effectively reducing the concentration of these persistent pollutants in contaminated environments [82,83].

The efficiency of xenobiotic degradation by microorganisms is influenced by several factors, including the physicochemical properties of the pollutant, the presence of specific enzymes, environmental conditions such as temperature, pH, and nutrient availability, and the metabolic versatility of the microbial community involved [78]. Recent advances in molecular biology and environmental biotechnology have opened new avenues for enhancing the microbial degradation of xenobiotics. A detailed understanding of these parameters and their interaction with microbial metabolism enables the design of effective bioremediation protocols tailored to specific contaminants and site conditions.

Techniques like metagenomics have allowed scientists to uncover novel genes and microbial species with previously unknown biodegradation capabilities, expanding the toolkit available for bioremediation efforts. Furthermore, synthetic biology approaches have enabled the engineering of microorganisms with tailored metabolic pathways that can degrade specific xenobiotics more efficiently, even under adverse environmental conditions [90]. These genetically engineered strains can be designed to possess heightened tolerance to toxic compounds, increased enzyme expression, and the ability to survive in diverse ecological niches, thereby boosting the overall effectiveness of bioremediation practices [91].

Understanding the degradation pathways of different xenobiotics is crucial for developing targeted bioremediation strategies. For instance, the breakdown of organophosphate pesticides involves hydrolysis by microbial enzymes like phosphotriesterases, leading to the release of harmless inorganic phosphate and simpler carbon compounds [92]. Similarly, polycyclic aromatic hydrocarbons (PAHs) are subjected to ring-cleaving dioxygenases that transform them into less complex aromatic acids, which are then further metabolized into non-toxic compounds [93]. These pathways highlight the microbial capacity to detoxify and mineralize xenobiotics, converting them into basic elements like carbon dioxide (CO_2_), methane (CH_4_), water (H_2_O), and biomass that integrate seamlessly into natural biogeochemical cycles.

## 7. Biodegradation of Environmental Pollutants

Environmental pollution has become a pressing global issue, significantly impacting both ecosystems and human health. Various pollutants, including heavy metals like lead, cadmium, arsenic, and mercury, as well as hydrocarbons and pesticides, pose severe threats to environmental integrity [94]. Heavy metals are particularly concerning because, unlike many organic pollutants, they are elemental and do not degrade into simpler, less toxic forms [95,96]. Their persistence in the environment can lead to bioaccumulation in living organisms, causing detrimental health effects and ecological imbalance.

Microbial processes play a pivotal role in mitigating the risks associated with heavy metal contamination through mechanisms such as biosorption, bioaccumulation, and biomineralization [72]. Microorganisms have evolved specialized strategies to interact with heavy metals, transforming these toxic elements into less harmful or immobilized forms. For instance, bacteria from the genera *Pseudomonas* and *Bacillus*, as well as fungal strains like *Saccharomyces*, demonstrate a remarkable ability to bind heavy metals to their cell walls. This binding process effectively sequesters the metals, converting them into stable forms that limit their mobility and bioavailability in the environment [72]. By reducing the concentration of free heavy metals in contaminated soil and water, these microorganisms help minimize the risks to plant and animal life and contribute to overall ecosystem health [95].

In addition to heavy metals, the biodegradation of hydrocarbons is a crucial aspect of environmental remediation efforts. Hydrocarbons are primarily released into the environment through oil spills and various industrial processes, leading to widespread contamination [97]. Certain bacteria, notably *Alcanivorax* and *Pseudomonas*, have developed the ability to utilize hydrocarbons as energy sources, effectively breaking down these compounds into less toxic substances like carbon dioxide and water [98]. This bioremediation process is essential for detoxifying environments impacted by oil spills, as it not only aids in the degradation of harmful pollutants, but also facilitates the restoration of affected ecosystems.

Similarly, the environmental persistence of pesticides, commonly used in agricultural practices, has raised concerns regarding soil and water pollution. Many pesticides are resistant to degradation, leading to their accumulation in the environment and potential entry into the food chain [99]. However, certain microorganisms, such as *Trichoderma* and *Bacillus*, possess specialized enzymatic pathways that effectively degrade these chemicals. By transforming pesticides into non-toxic byproducts, these microorganisms prevent their accumulation in the ecosystem and mitigate the risks posed to human health and wildlife [63].

The integration of microbial-based technologies with bioremediation strategies presents a holistic solution to combat various pollutants [79]. By harnessing the natural capabilities of microorganisms, it is possible to develop innovative remediation approaches that restore ecological balance and ensure cleaner soil and water resources. This multifaceted approach not only targets pollution, but also promotes the sustainability of ecosystems, paving the way to a healthier planet. Implementing such strategies can significantly enhance the resilience of ecosystems, allowing them to recover from the adverse effects of pollution while supporting biodiversity and human well-being [94]. Through the application of these natural processes, a reduction in environmental toxicity becomes feasible, contributing to the overarching goal of sustainable environmental management.

## 8. Microbial Management of Environmental Wastes

The microbial management of environmental waste is a fundamental component of ecological sustainability, leveraging the inherent capabilities of microorganisms to treat and recycle both organic and inorganic waste materials [100]. Naturally occurring microbes, including bacteria, fungi, and archaea, play a crucial role in the degradation of organic wastes such as food residues, agricultural by-products, and sewage sludge. These processes include fermentation, composting, and anaerobic digestion, which are essential for the effective breakdown of complex organic matter into simpler compounds [100,101].

The microbial breakdown of organic waste not only reduces its volume significantly, but also transforms it into valuable by-products [102]. For instance, anaerobic digestion leads to the production of biogas, a renewable energy source that can be utilized for heating or electricity generation, or as a vehicle fuel. Similarly, composting processes result in nutrient-rich compost that enhances soil fertility and structure, supporting sustainable agricultural practices [103]. By converting waste into beneficial resources, as illustrated in Figure 3, microbial management plays a vital role in diminishing the dependence on chemical fertilizers, which can degrade soil quality over time. This shift towards organic amendments also fosters a more environmentally friendly agricultural framework, contributing to soil health and crop productivity.

Recent advancements in biotechnology have further amplified the potential of microbial waste management. The development of genetically engineered strains with enhanced degradative capabilities represents a significant leap forward. For example, researchers have developed genetically modified bacteria that can break down plastic polymers, addressing one of the most pressing environmental issues: plastic pollution [104]. These engineered microorganisms can effectively metabolize plastic waste, converting it into harmless byproducts and reducing the accumulation of plastics in landfills and natural habitats.

In wastewater treatment facilities, diverse microbial consortia also play a pivotal role in the removal of organic and inorganic pollutants. These microbial communities are capable of metabolizing a wide range of contaminants, transforming them into non-toxic end products that can be safely reintroduced into the environment [105]. This bioremediation process not only purifies wastewater, but also minimizes the ecological impact of industrial discharges and urban runoff.

The integration of microbial solutions into waste management strategies promotes a circular economy, wherein waste materials are continually repurposed into valuable products rather than discarded. This innovative approach not only mitigates environmental pollution, but also fosters resource recovery, thereby contributing to a sustainable future [106]. Microbial management supports the principles of sustainability and resource efficiency by facilitating the reuse of materials and reducing waste generation. In addition to transforming waste into useful products, microbial management contributes to carbon sequestration and nutrient cycling within ecosystems [107]. Carrier engineering and microbial community regulation offer modern and easier approaches to enhancing the microbial degradation of various pollutants. Carrier engineering involves the application of biochar to immobilize and support microbial growth, improving pollutant mitigation [108]. On the other hand, the specific pollutant degradation goal can be achieved through microbial community regulation, along with optimization, interactions, and microbial activities [108]. By enhancing the organic matter content of soils through composting and by applying biogas digestate, microbial practices can improve soil structure and health, promoting biodiversity and resilience against climate change. Furthermore, this approach aligns with global sustainability goals by reducing greenhouse gas emissions associated with waste disposal and fostering environmentally responsible practices.

## 9. Conclusions

The identification of microbes with stress tolerance abilities, followed by their characterization, a compatibility interaction analysis, and the establishment of an efficient delivery system for utilization in crop plants, could be an interesting idea for sustainable agriculture. Various parameters, such as physiological, biochemical, ultrastructural, and molecular, can be assessed to understand plant–microbe interactions and the associated plant response mechanisms towards biotic and abiotic stresses. This study involves approaches such as genomics, proteomics, and metabolomics to develop an enhanced understanding of the metabolic pathways of microbial interactions with plants, including the production of enzymes, secondary metabolites, antioxidative compounds, and proteins for plant growth promotion through stress amelioration. The utilization of biological tools for stress amelioration would be a great asset for improving plant productivity. Presently, there is a great need for an in-depth study of such rhizospheric microbes and their mechanisms in stress amelioration so that crop productivity can be increased to meet the present food demand.

## Figures and Tables

**Figure 1 ijms-26-04515-f001:**
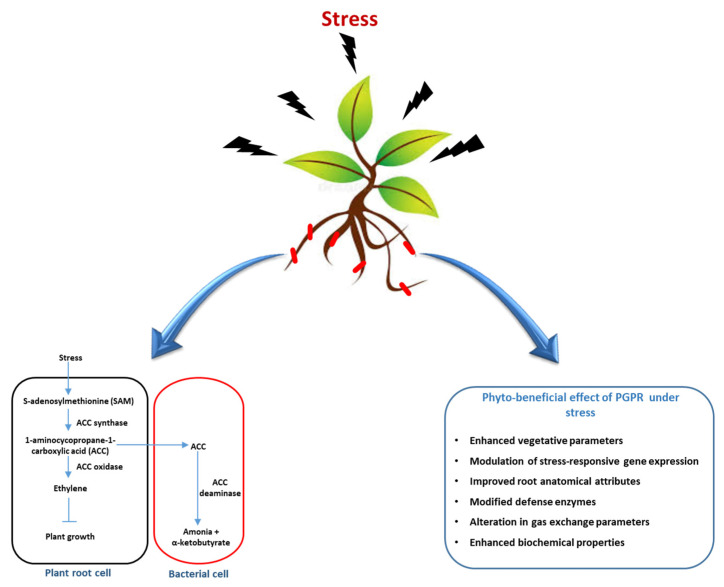
A schematic representation of the role of plant growth-promoting rhizobacteria (PGPRs) in mitigating plant stress through ACC deaminase activity, which lowers ethylene levels in stressed plants and leads to enhanced growth, improved root and biochemical attributes, the modulation of stress-responsive gene expression, and overall phyto-beneficial effects under adverse conditions.

**Figure 2 ijms-26-04515-f002:**
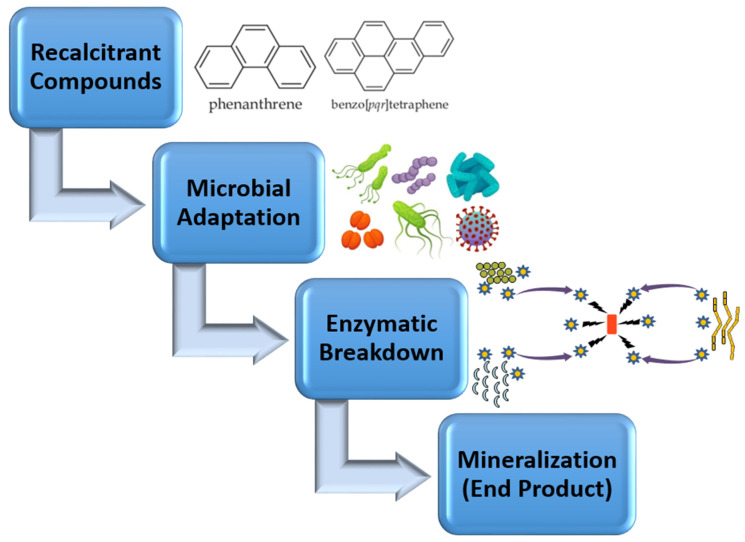
Stepwise microbial degradation of recalcitrant organic compounds. Recalcitrant compounds undergo microbial degradation, followed by enzymatic breakdown, ultimately leading to their mineralization into end products.

**Figure 3 ijms-26-04515-f003:**
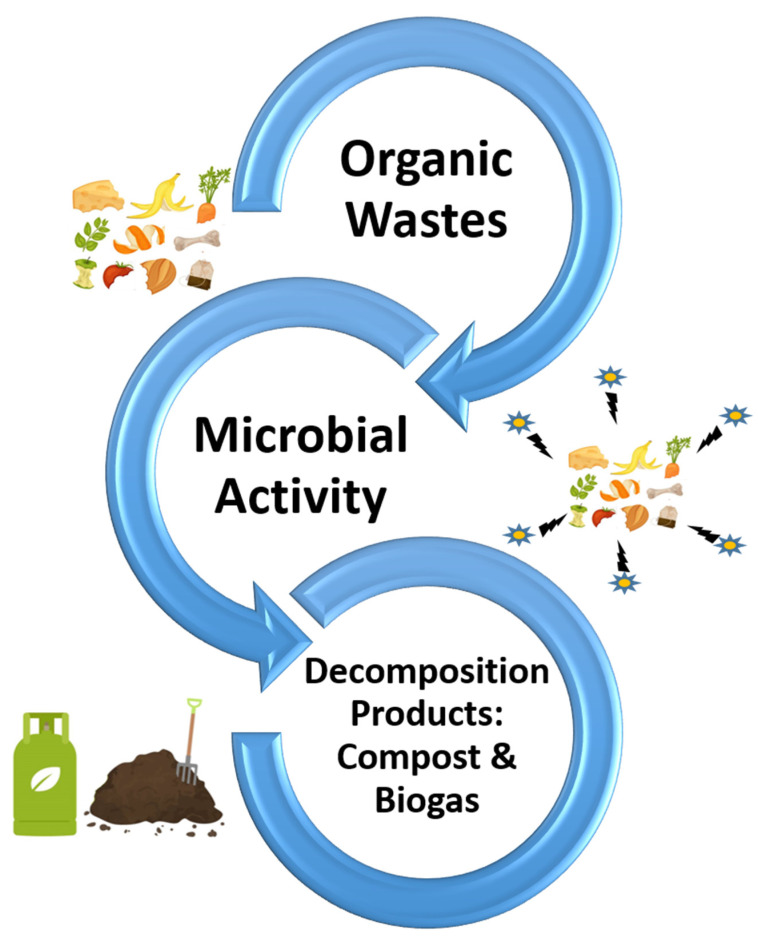
Microbial decomposition of organic wastes. Microbial activity converts organic wastes into valuable products such as compost and biogas, highlighting the sustainable recycling of organic matter.

**Table 1 ijms-26-04515-t001:** Microbial degradation of xenobiotic compounds.

S. No.	Xenobiotics	Microbes	Mechanism	Influencing Factors	Degradation Pathways	References
1	Polychlorinated Biphenyls (PCBs)	*Pseodomonas putida*	Reductive dichlorination and oxidation	pH, temperature	Anaerobic	[84,85]
2	Atrazine	*Anthrobacter* sp.	Hydrolysis followed by ring cleavage	Soil moisture, nitrogen content, organic matter	Aerobic	[86]
3	Polycyclic aromatic hydrocarbons (PAHs)	*Mycobacterium* sp.	Oxidation via deoxygenases enzymes	Oxygen levels, nutrient availability	Aerobic and anaerobic	[87]
4	Trichloroethylene (TCE)	*Burkholderia cepacia*	Reductive dichlorination through cometabolism	Microbial community structure	Anaerobic	[88]
5	Phenol	*Candida tropicalis*	Oxidative ring cleavage	pH, temperature, dissolved oxygen	Aerobic	[89]

## Data Availability

No datasets were generated or analyzed during the current study.

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
