# Peer review of "Exploring Microbial Ecosystem Services for Environmental Stress Amelioration: A Review"

_ijms, 2025, doi:10.3390/ijms26104515_

Round 1

Reviewer 1 Report

Comments and Suggestions for Authors

This is a nice review. However I think that in many parts it is a bit superficial, specially in the abstract, introduction, and conclusions. I made more specific comments in the attached file. If the authors agree to considerably change and improve these sections I believe that this manuscript has merit to be published.

Author Response

********************

Reviewer #1

********************

********************

Reviewer #2

********************

Comment: This manuscript systematically reviews the multiple mechanisms of microorganisms in mitigating environmental stresses such as drought, salinity, and pollutants, as well as their application potential. The theme closely aligns with the cutting-edge demands of sustainable agriculture and ecological restoration, and holds significant scientific significance and application value. The manuscript has a clear structure and coherent logic, covering multiple dimensions such as microbial diversity, stress response mechanisms, pollutant degradation and waste management, demonstrating strong comprehensiveness. Overall, this manuscript has the potential for publication after being revised in the following aspects.

Response: The authors thank the reviewer for acknowledging the study and providing valuable suggestions for significant improvements. Reviewer’s comments have been carefully addressed in the revised manuscript.

Comment: L 127-131: Some of the arguments lack specific experimental data support (for example, the case of Bacillus improving the salt tolerance of potatoes mentioned in Chapter 3 did not cite field trial data). It is recommended to supplement empirical studies on key microorganisms (such as crop yield after inoculation, changes in soil enzyme activity or metabolomics data).
Response: We accept the concern of the reviewer; the required information has been provided in the revised manuscript.

Comment: L 178-186: Chapter 4 mentions that "ACC deaminase reduces ethylene levels", but does not elaborate in detail the molecular association between the ethylene signaling pathway and the plant stress response. It is suggested to add schematic diagrams or discuss key regulatory genes.

Response: We accept the concern of the reviewer. The required figure has been provided in the revised manuscript.

Comment: Chapters 5-6 discuss "stubborn compounds" and "exotic substances" side by side, but there is an overlap in their definitions (for example, PAHs can be classified as both stubborn compounds and exotic substances). It is suggested to integrate it into a unified analysis of "refractory pollutants" or clarify the classification standards.

Response: Many thanks for the suggestion. Stubborn compounds (also called recalcitrant or persistent compounds) are chemicals that are resistant to natural degradation processes such as biodegradation, photodegradation, or hydrolysis and Exotic substances are unusual or anthropogenic (man-made) chemicals that are not naturally present in the environment or occur in very low concentrations. PAH is an overlap and therefore the suggested change has been made in the revised manuscript.

Comment: Chapter 8 does not discuss the practical challenges of microbial remediation (such as the colonization efficiency of strains, environmental adaptability, and the risk of horizontal gene transfer). It is suggested to supplement relevant restrictions and solutions (such as carrier engineering or microbial community regulation).

Response: The authors thank the reviewer for the suggestion. The required information has been provided in the revised manuscript.  

Comment: This is a nice review. However I think that in many parts it is a bit superficial, specially in the abstract, introduction, and conclusions. I made more specific comments in the attached file. If the authors agree to considerably change and improve these sections I believe that this manuscript has merit to be published.

Response: The authors thank the reviewer for appreciating the study and suggesting substantial corrections in the manuscript. We have addressed the reviewer’s comments in the revised manuscript.

Comment: I suggest to add in the title that this a Review so the readers can already get the context:

A review exploring....

or stress amelioration: a review

Response: The authors acknowledge the reviewer's concern. The title has been changed in the revised manuscript.

Comment: The abstract is too descriptive.

It is missing more about the objective of this review, as well as the structure of the paper and more highlights on key findings that this work provides and conclusions.

Therefore, I suggest to rewrite it.

Response: The authors agree with the reviewer’s comment. The abstract has been modified in the revised manuscript.

Comment: It is well structured.

The first paragraph can be more concise. The beginning is too superficial.

I suggest to split the second paragraph in two

Also, to add a last paragraph with the clear objectives of this review and how the review is structured.

Response: The suggested corrections have been incorporated in the revised manuscript.
Comment: Add references for these.

Response: The required information has been provided in the revised manuscript.

Comment: I think that section 2 is too long...

the focus of this review should be more on section 3. Therefore, I suggest the authors to shorten and make section 2 more concise. And to improve and give more information in Section 3.

Response: We agree with the reviewer’s comment. The required corrections have been made in the revised manuscript.

Comment: The font size and quality of the Figure must be improved.

Also, the legend should be more descriptive. The figure + legend should "stand alone".

Response: The authors agree with the reviewer’s comment. The required corrections have been made in the revised manuscript.

Comment: Fig.2: Same comments as in Figure 1.

Response: The authors agree with the reviewer’s comment. The required corrections have been made in the revised manuscript.

Reviewer 2 Report

Comments and Suggestions for Authors

This manuscript systematically reviews the multiple mechanisms of microorganisms in mitigating environmental stresses such as drought, salinity, and pollutants, as well as their application potential. The theme closely aligns with the cutting-edge demands of sustainable agriculture and ecological restoration, and holds significant scientific significance and application value. The manuscript has a clear structure and coherent logic, covering multiple dimensions such as microbial diversity, stress response mechanisms, pollutant degradation and waste management, demonstrating strong comprehensiveness. Overall, this manuscript has the potential for publication after being revised in the following aspects.

L 127-131: Some of the arguments lack specific experimental data support (for example, the case of Bacillus improving the salt tolerance of potatoes mentioned in Chapter 3 did not cite field trial data). It is recommended to supplement empirical studies on key microorganisms (such as crop yield after inoculation, changes in soil enzyme activity or metabolomics data).

L 178-186: Chapter 4 mentions that "ACC deaminase reduces ethylene levels", but does not elaborate in detail the molecular association between the ethylene signaling pathway and the plant stress response. It is suggested to add schematic diagrams or discuss key regulatory genes.

Chapters 5-6 discuss "stubborn compounds" and "exotic substances" side by side, but there is an overlap in their definitions (for example, PAHs can be classified as both stubborn compounds and exotic substances). It is suggested to integrate it into a unified analysis of "refractory pollutants" or clarify the classification standards.

Chapter 8 does not discuss the practical challenges of microbial remediation (such as the colonization efficiency of strains, environmental adaptability, and the risk of horizontal gene transfer). It is suggested to supplement relevant restrictions and solutions (such as carrier engineering or microbial community regulation).

Author Response

********************

Reviewer #1

********************

Comment: This is a nice review. However I think that in many parts it is a bit superficial, specially in the abstract, introduction, and conclusions. I made more specific comments in the attached file. If the authors agree to considerably change and improve these sections I believe that this manuscript has merit to be published.

Response: The authors thank the reviewer for appreciating the study and suggesting substantial corrections in the manuscript. We have addressed the reviewer’s comments in the revised manuscript.

Comment: I suggest to add in the title that this a Review so the readers can already get the context:

A review exploring....

or stress amelioration: a review

Response: The authors acknowledge the reviewer's concern. The title has been changed in the revised manuscript.

Comment: The abstract is too descriptive.

It is missing more about the objective of this review, as well as the structure of the paper and more highlights on key findings that this work provides and conclusions.

Therefore, I suggest to rewrite it.

Response: The authors agree with the reviewer’s comment. The abstract has been modified in the revised manuscript.

Comment: It is well structured.

The first paragraph can be more concise. The beginning is too superficial.

I suggest to split the second paragraph in two

Also, to add a last paragraph with the clear objectives of this review and how the review is structured.

Response: The suggested corrections have been incorporated in the revised manuscript.
Comment: Add references for these.

Response: The required information has been provided in the revised manuscript.

Comment: I think that section 2 is too long...

the focus of this review should be more on section 3. Therefore, I suggest the authors to shorten and make section 2 more concise. And to improve and give more information in Section 3.

Response: We agree with the reviewer’s comment. The required corrections have been made in the revised manuscript.

Comment: The font size and quality of the Figure must be improved.

Also, the legend should be more descriptive. The figure + legend should "stand alone".

Response: The authors agree with the reviewer’s comment. The required corrections have been made in the revised manuscript.

Comment: Fig.2: Same comments as in Figure 1.

Response: The authors agree with the reviewer’s comment. The required corrections have been made in the revised manuscript.

********************

Reviewer #2

********************

Comment: This manuscript systematically reviews the multiple mechanisms of microorganisms in mitigating environmental stresses such as drought, salinity, and pollutants, as well as their application potential. The theme closely aligns with the cutting-edge demands of sustainable agriculture and ecological restoration, and holds significant scientific significance and application value. The manuscript has a clear structure and coherent logic, covering multiple dimensions such as microbial diversity, stress response mechanisms, pollutant degradation and waste management, demonstrating strong comprehensiveness. Overall, this manuscript has the potential for publication after being revised in the following aspects.

Response: The authors thank the reviewer for acknowledging the study and providing valuable suggestions for significant improvements. Reviewer’s comments have been carefully addressed in the revised manuscript.

Comment: L 127-131: Some of the arguments lack specific experimental data support (for example, the case of Bacillus improving the salt tolerance of potatoes mentioned in Chapter 3 did not cite field trial data). It is recommended to supplement empirical studies on key microorganisms (such as crop yield after inoculation, changes in soil enzyme activity or metabolomics data).
Response: We accept the concern of the reviewer; the required information has been provided in the revised manuscript.

Comment: L 178-186: Chapter 4 mentions that "ACC deaminase reduces ethylene levels", but does not elaborate in detail the molecular association between the ethylene signaling pathway and the plant stress response. It is suggested to add schematic diagrams or discuss key regulatory genes.

Response: We accept the concern of the reviewer. The required figure has been provided in the revised manuscript.

Comment: Chapters 5-6 discuss "stubborn compounds" and "exotic substances" side by side, but there is an overlap in their definitions (for example, PAHs can be classified as both stubborn compounds and exotic substances). It is suggested to integrate it into a unified analysis of "refractory pollutants" or clarify the classification standards.

Response: Many thanks for the suggestion. Stubborn compounds (also called recalcitrant or persistent compounds) are chemicals that are resistant to natural degradation processes such as biodegradation, photodegradation, or hydrolysis and Exotic substances are unusual or anthropogenic (man-made) chemicals that are not naturally present in the environment or occur in very low concentrations. PAH is an overlap and therefore the suggested change has been made in the revised manuscript.

Comment: Chapter 8 does not discuss the practical challenges of microbial remediation (such as the colonization efficiency of strains, environmental adaptability, and the risk of horizontal gene transfer). It is suggested to supplement relevant restrictions and solutions (such as carrier engineering or microbial community regulation).

Response: The authors thank the reviewer for the suggestion. The required information has been provided in the revised manuscript.  

Round 2

Reviewer 1 Report

Comments and Suggestions for Authors

The authors addressed my suggestions. Therefore, I recommend this article to be accepted.

Reviewer 2 Report

Comments and Suggestions for Authors The authors have modified the manuscript earnestly, I think the revised manuscript is acceptable.